# The cost of aging: Economic growth perspectives for Europe

Thaveesha Jayawardhana[1], Ruwan Jayathilaka[2]*, Thamasha Nimnadi[1], Sachini Anuththara[1], Ridhmi Karadanaarachchi[1], Kethaka Galappaththi[2], Thanuja Dharmasena[3]

**1** SLIIT Business School, Sri Lanka Institute of Information Technology, Malabe, Sri Lanka, **2** Department of Information Management, SLIIT Business School, Sri Lanka Institute of Information Technology, Malabe, Sri Lanka, **3** Gender and Environment Advisor & Acting Communications Officer, UN-Habitat, Colombo, Sri Lanka

\* ruwan.j@sliit.lk

## Abstract

This study explores the causal relationship between the economy and the elderly population in 15 European countries. The economy was measured by the Per Capita Gross Domestic Product growth rate, while the population aged above 65 as a percentage of the total was considered the elderly population. The data were obtained from a time series dataset published by the World Bank for six decades from 1961 to 2021. The Granger causality test was employed in the study to analyse the impact between the economy and the elderly population. An alternate approach, wavelet coherence, was used to demonstrate the changes to the relationship between the two variables in Europe over the 60 years. The findings from the Granger causality test indicate a unidirectional Granger causality from the economy to the elderly population for Luxembourg, Austria, Denmark, Spain, and Sweden, while vice versa for Greece and the United Kingdom. Furthermore, for Belgium, Finland, France, Italy, Netherlands, Norway, Portugal, and Turkey, Granger causality does not exist between the said variables. Moreover, wavelet coherence analysis depicts that for Europe, the elderly population negatively affected the economic growth in the 1960s, and vice versa in the 1980s.

## Introduction

Economists have debated the connection between the ageing population and economic growth for an extended period. Researchers have disagreed about the relationship between the ageing population and economic expansion. Governments and professionals worldwide are interested in the impact of the former on economic growth, however, different age groups have varying degrees of productivity and, financial needs [1]. The number of persons aged 65 and above was expected to rise gradually over the world from 1960 to 2020. This number rose from 150 million in 1960 to 722 million in 2020 around the world. European countries are currently experiencing an increase in ageing population due to rising longevity and falling fertility rates. The population's share of people over 65 is surging more than any other age group. This process presents significant financial, social, and economic challenges, especially at the

from World Bank Indicators (https://data.worldbank.org/indicator) datasets.

**Funding:** The authors received no specific funding for this work.

**Competing interests:** The authors have declared that no competing interests exist.

macroeconomic level. With an ageing population, concerns arise about increasing public spending on long-term care, healthcare, and other age-related expenses, particularly through the pension system [2–7]. European "active ageing" policy, which has two complementary goals of raising the retirement age and increasing the employment rate of older employees, strategies to solve the ageing population dilemma.

However, with the -challenges -in a post-pandemic environment, such as political divisiveness, one can argue that -economic progress is sluggishly. According to the publications of the European Commission [8, 9], the changes in population growth, the contribution to the economy and the overall lag in development could be major issues debilitating the growth momentum in Europe. Empirical evidence shows that Europe would be one of the continents facing the actual cost of ageing [10]. The same report highlighted how the ageing population could burden society with high health care costs, which can markedly harm Gross Domestic Product (GDP) growth in European countries [11]. Therefore, the report pointed out that these ageing-related costs need to be addressed in the first instance.

A significant rise in ageing population in Europe is expected, with the current estimated percentage, i.e., 19% is forecasted to rise up to 29% by 2070 [12]. Thus, it indicates that Europe has to plan ahead on the strategy on how well the tasks can be rolled out in the long run. Europe has allocated funds exceeding United States Dollars (USD) 3.7 billion (Bn) on new investments to implement the Europe-based ageing care plan, which is an significant step towards a sustainable solution. Based on the data available in Europe since 2015, a birth rate decline is apparent [11]. In another study, it was noted that the impact of the ageing population affect the GDP growth in the adjoining European countries [10]. The data was assessed using Granger Causality tests.

The principal contribution of this study is to examine the relationship between economic growth and population ageing in European countries. To the best of the authors' knowledge, this study has been the first to investigate the relationship between the elderly population and economic growth for 15 European nations, using time series data and a Granger causality test and coherence analysis.

As such, the present research differs from the existing studies and fills a gap in the literature that in many ways. Firstly, this study employs the latest data amassed over a considerable period and spans several nations in the European continent. A very few studies of this nature have been undertaken for Europe even for shorter time periods. To the best of the researchers' knowledge, thus far, studies have not focused on this topic in this magnitude across a lengthy time period. Therefore, this study is unique for its comprehensiveness, and how extensively the countries and years are covered. Secondly, the country-wise analysis allowed the researchers to examine how the two variables behave in various European nations. As such, the study findings offer important insights into how well recent programmes in various countries have worked. In doing so, this study fills a gap in the existing knowledge on the ageing population and economic growth in Europe. Thirdly, while separate European studies have examined how the ageing population affects the economy and vice versa, this study examines the impact of two variables on each other simultaneously. In addition, this research contributes to filling the empirical gap of capturing the Granger causality concerning the relationship between the elderly population and economic growth in the European context in a single study. Moreover, to the authors' knowledge, the wavelet technique has not been used in previous research to analyse the relationship between Per Capita Gross Domestic Product (PGDP) and the elderly population. Furthermore, using the wavelet technique for analysis in this research area would produce precise and reliable data for future research and decision-making.

The remainder of the paper is organised as follows. The main section discusses the problem in detail, while the second section explores the theoretical background of the variables studied.

The third section introduces the data and methodology, and the fourth section presents the empirical findings and interpretations relating to the research problem. The fifth section brings the research to a close by making recommendations and identifying policy implications.

## Literature review

For several decades, countries in Europe have seen an increase in elderly population. For example, in the European Union (EU), the old-age dependency ratio (the ratio of the population aged 65 or over to the population aged 15–64) nearly doubled from 15.2% in 1960 to 29.9% in 2016. Maintaining fiscal sustainability will be extremely difficult in the face of such a significant demographic shift [13].

The relationship between the elderly population and economic growth has been a much-debated issue in global research. Several leading institutions, including the United Nations, International Monetary Fund, and European Union (EU), were keen to investigate the nature of its relationship. This is mainly because the elderly population has a significant impact on future policy implementation [9, 11, 14]. Some studies argue against the popular belief in prevailing in research that the elderly population negatively leads to economic growth [15–17]. It is essential to set strong benchmarks for the ageing population to evaluate concerns in this regard. In Europe, as in many other countries, adults above 65 years of age are the population's ageing cohort [18]. The estimated ageing population growth for Europe has exceeded 20% in the past three years and is expected to further increase.

Further to the discussions on how the ageing population may impact, Europe has proposed strong government backed policies to safeguard the ageing population on health benefits and guaranteeing dignity [19]. In Italy, Greece and Turkey, ageing population dominates the tourism industry with many elders taking up travel [20]. This finding has not been analysed in-depth on how the elderly indirectly contribute to GDP growth.

Specifically, in terms of deploying the ageing population as a growth catalyst, decreasing dependency through introducing an early health intervention strategy is a must. Especially Finland and Denmark have seen more than 30% of their population ageing, pushing the health care costs slightly higher. It is also highlighted that the ageing population can strongly influence the working population, exerting much strain on public budgets when funds are to diverted to elderly care compromising the funds allocated for the development of education and health in a more generalisable context [21, 22]. Furthermore, during the last decade, all Nordic nations experienced an increase in the older age groups [23]. In terms of overall development, the rising elderly population exerts a burden on the working population and has significantly changed the migration policy of these countries [21]. Post COVID-19 pandemic, these countries have employed young immigrants to the workforce specially from developing countries, benefiting both developed and developing countries [24].

Europe has wellness-based strategies in place in terms of lifestyle of the residents, particularly prioritising productivity as opposed to longer unproductive work time, e.g., countries such as Italy and France prioritising two-hour lunch breaks. Thus, they focus more on reducing the pressure of productivity and focusing on mental health. This social context is highly important in understanding that Europe has a strong long-term strategy that aligns with its identity, as the region fosters cultural awareness. The egalitarian context of Europe provides an equity-based opportunity for everyone, which is strong adherence to managing the challenges of the ageing population [21].

In countries such as Italy, Greece, and France, the impact of the ageing population is quite low compared to other European countries [25]. This is because these nations place a strong

emphasis on the culture for optimal engagement of the aged population. After retirement, many of the aged population serve in hotels, restaurants and primarily in establishments in the hospitality trade. These opportunities enable elderly individuals to feel economically active, which can offset any health-related costs they may incur to society.

However, reference to the impact of aged population, Finland has reported the highest impact on their GDP growth. During 2019–2021, a 6% sharp decline was evident in the country's productivity due to the aged population labour participation being lowered [9], and the ageing population is a concerning factor that probably impact the country's growth for the next three to five years if the birth rates tend to decline.

The global elderly population is growing as health and treatment standards improve, a phenomenon known as the "demographic revolution." According to WHO, the global the elderly population is expected to reach 727 million by 2020 and more than 1.5 billion by 2050 [26]. The "demographic transition" initially resulted in unprecedented global population growth in the early 1970s. In general, rapid population growth contributed to swift increase in the number of workers, and fast increase in the number of workers contributed to speedy increase in GDP. As the demographic transition ends, the population is expected to grow slower and, after many decades, to reach a significantly high peak than currently [27]. Globally, an unprecedented shift in population demographics is taking place: for the first time, older people outnumber children. As the average age of the world's population rises, it not only poses problems for social welfare but also poses a significant challenge for energy systems and efforts to combat climate change. Europe, like the rest of the world, is undergoing a rapid demographic shift. Consistently low birth rates and higher life expectancy can explain the shift in the shape of the European Union's age pyramid [28].

Additionally, factors like the environment, transportation, and technology have an impact on how the economy is growing [29–35]. These factors are also influenced by the aging population. From the standpoint of public spending, when the relationship between economic growth and the environment is examined, it is emphasised that for both economic growth and a cleaner environment, governments should have balanced and sustainable public spending policies. Participation in environmental activities by the elderly population will boost economic growth to the extent that they value these factors. Economies that want to gain a competitive edge can raise their welfare level and achieve their goals in becoming information societies. Information technology has a short-term positive impact and a long-term negative impact on Turkey's economic growth [36]. By making the elderly population more familiar with technology and engaged in productive work, the country's economy might grow.

Furthermore, when there is less risk and more safety protocols are in place, people who are concerned about public health would be kept at ease to attend the events. Event managers could benefit from a number of initiatives aimed at allaying consumer worries and fostering trust during the event in order to promote a secure and healthy atmosphere and encourage event participation [37]. Thereby, the mental health of elderly population will improve encouraging them to contribute more to the economy.

The importance of the ageing population and related policymaking has become a maxim for many European countries [18]. The lack of government push to develop proper aged care systems has been a primary issue [38]. A key action towards addressing the issue of the burden of higher percentage of elderly population can be initiated by illustrating the importance of new births to the economy and improving the incentives for starting families. These would highlight the importance of family, religion, culture and society.

## Data and methodology

The study' main secondary data source was the online version of World Bank Open Data [39]. Owing to the unavailability of data sets for long timespans for some countries, this study was restricted to the 1961–2021 period, where data were readily available. The countries studied were also limited to fifteen (15), including Austria, Belgium, Denmark, Finland, France, Greece, Italy, Luxembourg, Netherlands, Norway, Portugal, Spain, Sweden, Turkey, and the United Kingdom. Economic growth was measured by each nation's PGDP growth rate (as an annual percentage) using constant local currency, allowing cross-country comparisons. In addition, the elderly population was measured using the population aged 65 and older (as a percentage of the total population). The elderly population is defined as people aged 65 and over [40]. The data file used for the study is presented in S1 Appendix.

The methodology of the study is based on Granger Causality analysis. The following equations are estimated to test the direction of causality from the elderly population to economic growth Eq (1) and from economic growth to the elderly population Eq (2). Applicability of the Granger causality test prevails in the study since its major assumption (i.e., two variables are stationary) is not violated.

$$PGDP_t \; = \; C_0 \; + \; \sum_{k=1}^{p} \alpha_i \, PGDP_{t-k} \; + \; \sum_{k=0}^{p} \beta_i \, EPOP_{t-k} \; + \; \varepsilon_t \tag{1}$$

$$EPOP_t \; = \; Z_0 \; + \; \sum_{k=1}^{p} \gamma_i \, EPOP_{t-k} \; + \; \sum_{k=0}^{p} \sigma_i \, PGDP_{t-k} \; + \; \varepsilon_t \tag{2}$$

Where index $k$ refers to the country ($k = 1 \dots$ N), $t$ to the time period (t = 1 $\dots$ T), $p$ to the maximum lag. The autoregressive parameters in Eq (1) and in Eq (2) and the regression parameters' slopes in Eq (1) and in Eq (2) are constant. It is also assumed that parameters in Eq (1) and in Eq (2) are identical for all countries, whereas the regression parameters in Eq (1) and in Eq (2) could have an individual dimension. Two stationary covariance variables, X and Y, are tracked over t periods. If it can be demonstrated that lagged values of a variable X have a significant influence on a regression model of Y that depends not only on X but also on its own lagged values Yt-1, Yt-2. Therefore, it can be argued that X Granger causes Y and potentially Y Granger causes changes in X [41].

Afterwards, the wavelet coherence approach was used to visually complement the results obtained from the panel Granger causality test.

$$\psi^{a,b}(x) \; = \; |a|^{-\frac{1}{2}} \psi \left( \frac{x-b}{a} \right) \tag{3}$$

The function can denote wavelet $\psi^{a,b}(x)$ with $a$ as the contraction and $b$ as the translation in Eq (3). Furthermore, $a$ is the scaling factor that determines the compression level of the wavelet [42]. When the scale factor is small, the wavelet is more compressed. The shifting of a wavelet, in the means of delaying or advancing, is represented by $b$. Another advantage of wavelet coherence is the ability to understand the relationship direction and leading variable during different time lags. Time series is divided into a frequency-time domain under the wavelet coherence concept developed by Goupillaud and Grossmann [43]. Several research studies later enhanced the applicability of wavelet in social sciences [44–47]. In-depth understanding was made possible with the combination of the Granger causality test and the wavelet coherence approach.

## Result and discussion

The key findings of the empirical approach are presented in this section. First, the descriptive statistics of the data are presented in Table 1. During the period under the study, the average PGDP growth rate and elderly population growth rate were 2.26 and 14.19, respectively. Among the 15 countries considered in this study, Spain has the lowest PGDP value and Portugal has the highest value. Moreover, Italy has the highest elderly population and Turkey has the lowest elderly population.

The flow of PGDP growth rates for the 15 European countries is presented in Fig 1 for the period of 1961–2021. Overall, the PGDP growth rates of 15 countries indicate rapid changes during this period. According to Fig 1A, Italy's economy expanded throughout the first half of the 1960s, consequent to the Marshall Plan and the launching of new industries, but after 1966, it started to decline relative to other nations. This change is attributable to political, economic, and social issues. Fig 1A and 1B show that all 15 European countries considered have

**Table 1. Descriptive statistics.**

| Country | Variables | Mean | SD | Min | Max |
|---|---|---|---|---|---|
| Austria | PGDP growth rate | 2.212293 | 2.28395 | -7.12093 | 5.950497 |
| | EPOP | 15.57218 | 1.874051 | 12.33807 | 19.47894 |
| Belgium | PGDP growth rate | 2.16839 | 2.260238 | -6.13223 | 6.328284 |
| | EPOP | 15.51743 | 2.041466 | 12.02186 | 19.57415 |
| Denmark | PGDP growth rate | 1.969686 | 2.326 | -5.41401 | 8.400119 |
| | EPOP | 15.06006 | 2.418148 | 10.76309 | 20.34229 |
| Finland | PGDP growth rate | 2.372987 | 3.073409 | -8.51303 | 9.656904 |
| | EPOP | 13.91189 | 4.126259 | 7.454385 | 22.95518 |
| France | PGDP growth rate | 2.062852 | 2.3511 | -8.03438 | 6.774307 |
| | EPOP | 15.08924 | 2.458967 | 11.72142 | 21.08683 |
| Greece | PGDP growth rate | 2.299104 | 4.796017 | -10.0163 | 12.3098 |
| | EPOP | 14.58759 | 4.34845 | 7.165319 | 22.64105 |
| Italy | PGDP growth rate | 1.995462 | 2.961322 | -8.59787 | 7.486419 |
| | EPOP | 15.92397 | 4.275948 | 9.653221 | 23.60768 |
| Luxembourg | PGDP growth rate | 2.491514 | 3.492111 | -7.58648 | 10.54712 |
| | EPOP | 13.4555 | 0.90272 | 10.96928 | 14.64728 |
| Netherlands | PGDP growth rate | 2.046686 | 2.236718 | -4.33159 | 7.161982 |
| | EPOP | 13.12623 | 2.976826 | 9.015998 | 20.45085 |
| Norway | PGDP growth rate | 2.339383 | 1.974516 | -2.95859 | 5.753542 |
| | EPOP | 14.84687 | 1.608181 | 11.21001 | 17.78928 |
| Portugal | PGDP growth rate | 2.836051 | 3.926486 | -8.53865 | 13.61509 |
| | EPOP | 14.39347 | 4.317948 | 8.154703 | 23.14899 |
| Spain | PGDP growth rate | 2.444509 | 3.359815 | -11.2533 | 10.80449 |
| | EPOP | 13.79762 | 3.651362 | 8.311815 | 20.31885 |
| Sweden | PGDP growth rate | 1.975999 | 2.283209 | -5.15126 | 6.292338 |
| | EPOP | 16.72426 | 2.251117 | 11.93791 | 20.4697 |
| Turkey | PGDP growth rate | 2.839001 | 3.89511 | -7.14785 | 10.06785 |
| | EPOP | 5.530154 | 1.598789 | 3.29115 | 9.285825 |
| United Kingdom | PGDP growth rate | 1.90563 | 2.638312 | -9.60134 | 7.049388 |
| | EPOP | 15.35616 | 1.820833 | 11.8618 | 18.8472 |

Note: PGDP denotes the Per Capita Gross Domestic Product, EPOP denotes the Elderly population. The table is based on 61 observations.

Source: Authors' illustrations based on STATA software

suffered effects of economic recession due to high interest rates, unemployment growth etc. [48]. In general, the economy of any country can fluctuate, which can be affected by various factors such as high inflation, oil crisis, financial crisis etc. In Europe, the 1973 oil crisis, stagflation, the decline of traditional industries, and inefficient production contributed to wage

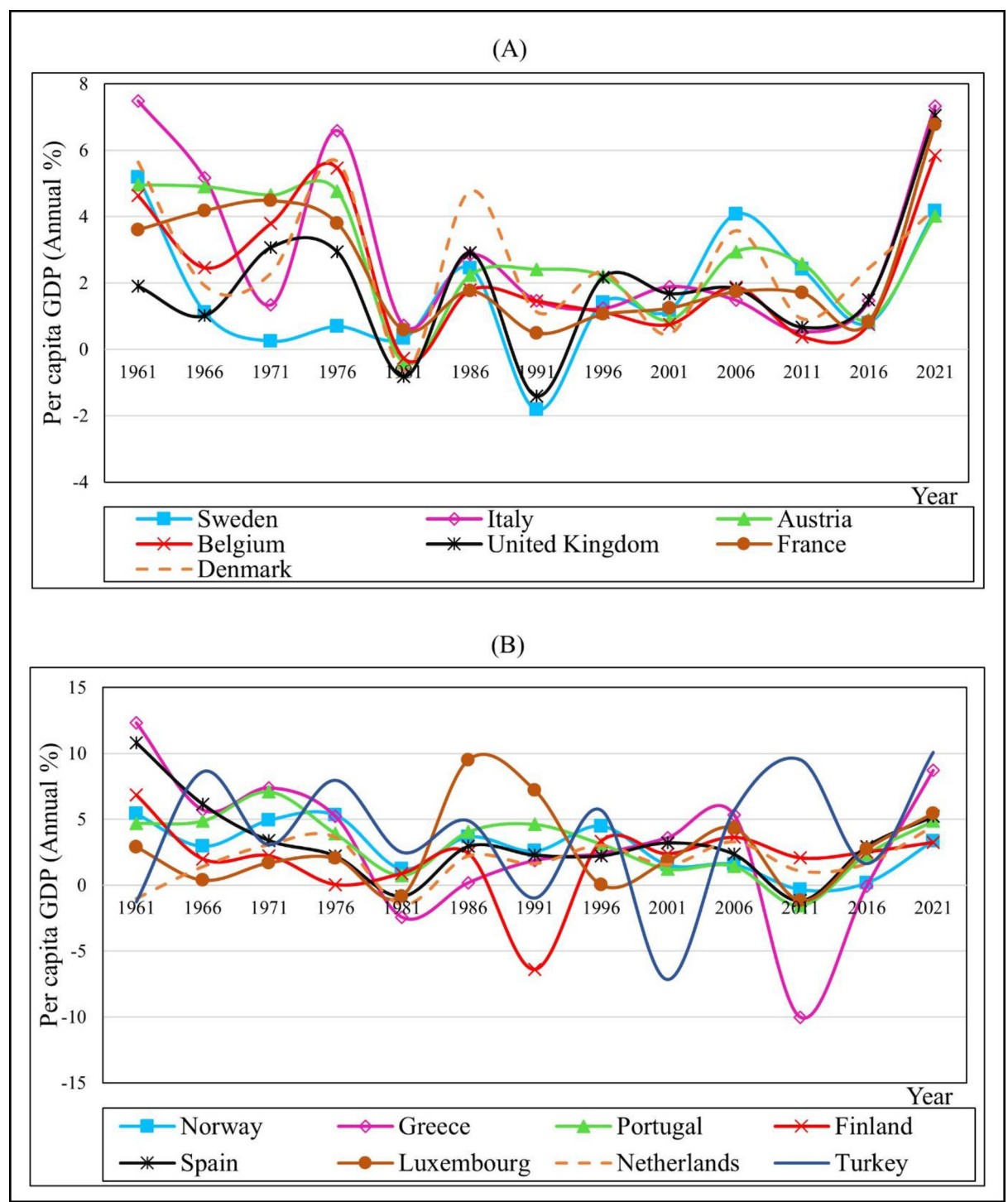

**Fig 1. PGDP growth (annual percentage): A comparison.** Source: Based on WDI [39].

disputes [49]. Early 1980s recession, deflationary government policies such as spending cuts, use of monetarism to reduce inflation, and transition from a manufacturing to a service economy were also among key influencing factors. The United Kingdom and Sweden in Fig 1A and Norway in Fig 1B were in severe recession in 1991. Additionally, Turkey's PGDP was rising in 2011, while Greece's PGDP was substantially lower (Fig 1B).

The 15 European countries with the highest and lowest mean values on old population growth rates are shown in Fig 2 in line with the flow of the elderly population over 65 between 1961 and 2021. In all countries, the elderly population slowly grew throughout these 60 years. Fig 2A shows that Italy has the highest average elderly population growth rates from 2000 to 2021, indicating the largest rise than any country. Additionally, Fig 2B depicts Turkey's low elderly population share pattern throughout the 61 years compared to all the other nations.

## Unit root test

Researchers examined the stationarity of the elderly population and the PGDP growth rate before performing the Granger causality analysis. Accordingly, the study used the Dicky Fuller (Dfuller) unit root test [50] and the Phillips Perron (Pperron) unit root test [51], created by David Dickey and Wayne Fuller, and Peter C. B. Phillips and Pierre Perron, respectively. The null hypothesis of the Dfuller test is that there is a unit root, which implies that the data series is not stationary. The alternative hypothesis may differ depending on the test version used, but it is often stationarity or trend stationarity. The Pperron test indicates a null hypothesis, meaning that the variable has a unit root; the alternative is that a stationary process developed the variable. The time series unit root test results from the Dfuller test for the increase in PGDP and the proportion of the population over the age of 65 are shown in Table 2 (annual percentage). Dfuller drift test results are shown in S2 Appendix and Dfuller trend test results are shown in S3 Appendix. Table 3 displays the outcomes of the Pperron time series unit root test.

A Dfuller unit root test was performed first on the PGDP growth rate variable. The first level unit root of the PGDP variable was performed for all 15 European nations. The same test was applied to the elderly population variable, and the researchers discovered that Luxembourg was unit root at the elderly population variable's first level (EPOP), and all 14 of the remaining nations became stationary at the second difference (DDEPOP).

The Pperron unit root test was then applied to the PGDP growth rate variable. The PGDP variable was stationary at the first level for all 15 nations, comparable to the Dfuller test. Following that, the elderly population variable was put through the Pperron test, which revealed that Luxembourg had unit roots at the elderly population's first level (EPOP), which is similar to the Dfuller test. The second difference in the elderly population led the remaining 14 countries to become stationary (DDEPOP).

After estimating the stationary test, the stability condition was checked. The resulting graph (Fig 3) of eigenvalues above confirms that the estimate is stable. The stability of the long-run coefficient estimation results demonstrates that the model is suitable for Granger causality functions. Once stability is achieved, the next stage of the analysis is continued. It depicts the eigenvalue stability condition by plotting all the companion matrix's real and imaginary eigenvalues. The stability condition is satisfied because the roots of the companion matrix are all contained within the unit circle when the Vector Auto Regression (VAR) model is used [52, 53]. In addition, the stability table is attached in S4 Appendix. The stability condition is satisfied if the table's modulus value is less than 1.

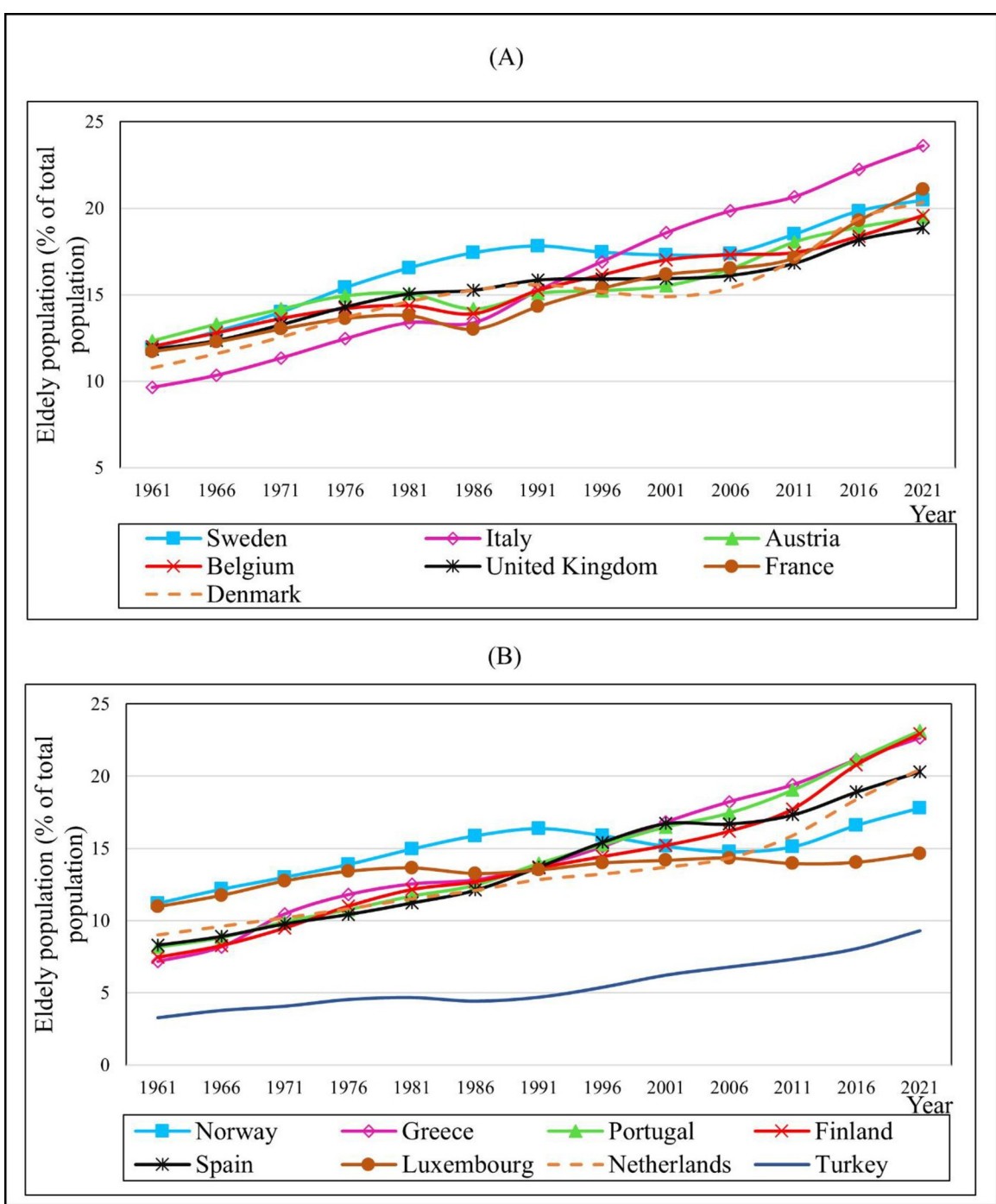

**Fig 2. Population aged 65 and older: A comparison.** Source: Based on WDI [39].

## Lag length criteria

The next step is to determine the optimal lag length of the VAR specification. Lag selection criteria report the final prediction error (FPE), Akaike's information criterion (AIC), Schwarz's Bayesian information criterion (SBIC), and the Hannan and Quinn information criterion (HQIC) on lag order selection statistics. The optimal lag length is determined based on AIC,

**Table 2. Dfuller test result.**

| Country | PGDP | EPOP | DEPOP | DDEPOP |
|---|---|---|---|---|
| Austria | -6.095*** | 0.9960 | -1.5830 | -3.712*** |
| Belgium | -6.493*** | 1.0130 | -1.4200 | -4.361*** |
| Denmark | -6.53*** | 1.5160 | -1.0990 | -4.22*** |
| Finland | -5.01*** | 5.3750 | -1.0230 | -3.754*** |
| France | -5.681*** | 4.7210 | -1.1660 | -3.231** |
| Greece | -4.948*** | 1.2270 | -2.3360 | -7.646*** |
| Italy | -5.821*** | 2.7320 | -1.7260 | -3.266** |
| Luxembourg | -5.61*** | -4.889*** | | |
| Netherlands | -5.411*** | 9.7330 | 0.1080 | -3.874*** |
| Norway | -4.161*** | -1.7130 | -0.6050 | -3.966*** |
| Portugal | -4.867*** | 10.2000 | -1.1740 | -4.13*** |
| Spain | -4.956*** | 1.9910 | -1.2970 | -4.104*** |
| Sweden | -5.783*** | -2.5370 | -1.0790 | -5.295*** |
| Turkey | -7.589*** | 6.4380 | 1.0790 | -2.642* |
| United Kingdom | -6.786*** | -0.1980 | -1.5500 | -5.055*** |

Source: Authors' illustrations based on STATA software.

SBIC, and HQIC. Empirical evidence has shown that the optimal lag length should be the one that minimises the SBIC, AIC, and HQIC information criteria [54]. When mixed results are obtained, the decision is based on the minimum value in the AIC criterion. According to S5 Appendix, in Belgium, Denmark, Finland, France, Italy, Netherlands, Norway, Portugal, Spain, and Turkey, the AIC, HQIC, and SBIC criteria are lower when selecting one lag. Selecting zero lags for Greece and two lags for Luxembourg, lowering the AIC, HQIC, and SBIC criteria. Further, in Austria, Sweden, and the United Kingdom, the AIC criterion is lower when selecting six lags.

**Table 3. Pperron test results.**

| Country | PGDP | EPOP | DEPOP | DDEPOP |
|---|---|---|---|---|
| Austria | -6.177*** | 0.1960 | -2.3350 | -3.877*** |
| Belgium | -6.617*** | 0.3030 | -2.2440 | -4.451*** |
| Denmark | -6.535*** | 0.4460 | -1.5860 | -4.239*** |
| Finland | -4.955*** | 2.7110 | -1.6220 | -3.881*** |
| France | -5.779*** | 2.3250 | -1.9940 | -3.521*** |
| Greece | -5.017*** | 0.5700 | -2.4930 | -7.646*** |
| Italy | -5.869*** | 1.4190 | -2.5500 | -3.617*** |
| Luxembourg | -5.647*** | -3.27** | | |
| Netherlands | -5.5520*** | 4.9830 | -0.5700 | -3.9520*** |
| Norway | -4.1460*** | -1.3750 | -1.1600 | -3.9860*** |
| Portugal | -4.9180*** | 5.6360 | -1.6030 | -4.2520*** |
| Spain | -4.8800*** | 0.9250 | -2.0050 | -4.1940*** |
| Sweden | -5.69*** | -1.5950 | -1.4600 | -5.369*** |
| Turkey | -7.583*** | 3.2790 | -0.3250 | -2.678* |
| United Kingdom | -6.775*** | -0.3550 | -2.0520 | -5.1*** |

Source: Authors' illustrations based on STATA software.

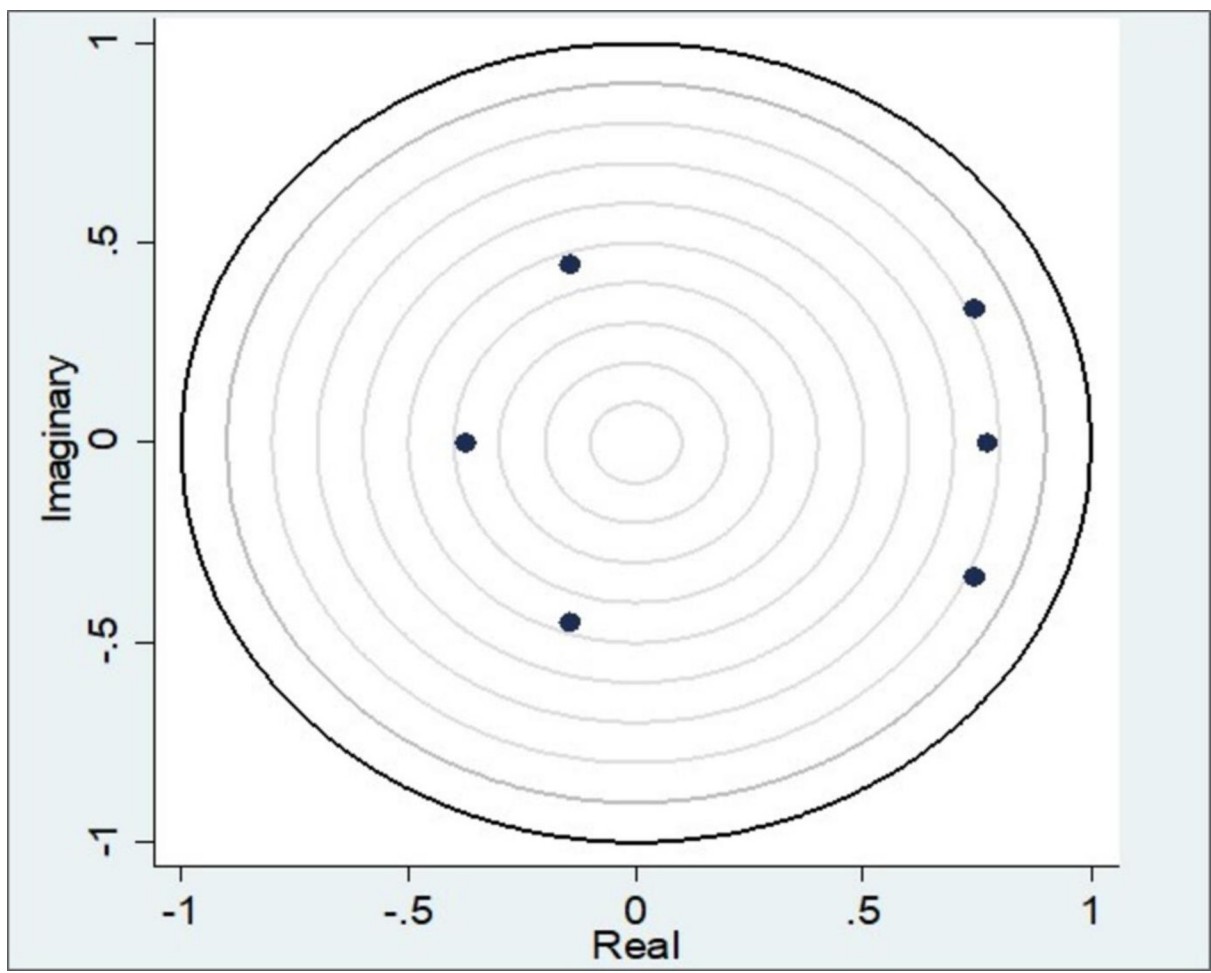

**Fig 3. Roots of the companion matrix.** Source: Authors' illustrations based on STATA software.

## VAR estimation

VAR model is a multivariate time series model that relates current observations of a variable to past observations of itself and other variables in the system. The coefficients from the VAR model instruments for the elderly population and economic growth are presented in S6 Appendix. In this model, all variables are in the first level for PGDP and for Luxembourg EPOP in first differences while other countries' EPOP in second differences are treated as endogenous. The results suggest the existence of significant simultaneous interdependencies between the elderly population and economic growth.

VAR models can be used to investigate relationships between the variables because these describe the joint generation process of several variables over time. One type of relationship between time series is Granger causality. Granger causality is an effective method for describing the interdependence of time series in reduced-form VARs. Granger causality test statistics are frequently used to determine whether lagged values of one variable contribute to the prediction of another variable [55].

## Granger causality test

The contribution of individual countries is tested to ensure the existence of Granger causality. For this purpose, we estimate Eqs (1) and (2) which differ among countries in our dataset and the hypothesis is tested individually for each country between PGDP and elderly population growth rate. Granger causality has the advantage of allowing a researcher to pinpoint directional influences of regions on one another without requiring any prior assumptions about which regions are involved in specific subnetworks. This is due to the fact that subnetworks are identified using data [56]. The results of the Granger causality test are presented in Table 4, with no bidirectional causality relation observed for any country and no causality relation observed for eight countries (Belgium, Finland, France, Italy, Netherlands, Norway, Portugal, Turkey) among the 15 countries in our dataset. Moreover, one-way causality relation is observed for seven countries: from elderly population growth rate to PGDP for two (Greece and the United Kingdom) and in the opposite direction for five (Luxembourg, Austria, Denmark, Spain, and Sweden). In Austria, Denmark, Spain, and Sweden, the current PGDP level are useful in forecasting the second difference in the population share of the elderly and in Greece and the United Kingdom second difference in the population share of the elderly is useful in forecasting the current PGDP level. The decrease in the number of countries having bidirectional Granger-causality is perhaps due to some health factors, pension planning, insurance planning etc. It nonetheless shows one-way Granger-causality for 50% of sample countries.

This study illustrates that at least 50% of the sample has a medium-level impact on the country's economic growth by the rising number of ageing populations. These changes would probably impact the governments and open market policies in the economies. When

**Table 4. Granger causality test results.**

| Country | PGDP → EPOP | EPOP→PGDP | PGDP → EPOP |
|---|---|---|---|
| Luxembourg | 8.6609** | 1.9029 | one-way (→) |
|  | **PGDP → DDEPOP** | **DDEPOP →PGDP** | **PGDP → DDEPOP** |
| Austria | 6.4806* | 1.5392 | one-way (→) |
| Belgium | 4.1877 | 4.6542 | no-way (↔) |
| Denmark | 8.7784** | 1.1243 | one-way (→) |
| Finland | 1.0453 | 2.0922 | no-way (↔) |
| France | 4.7306 | 3.4062 | no-way (↔) |
| Greece | 2.2245 | 13.109*** | one-way (←) |
| Italy | 3.1479 | 2.2305 | no-way (↔) |
| Netherlands | 0.23701 | 2.1756 | no-way (↔) |
| Norway | 2.8655 | 2.5513 | no-way (↔) |
| Portugal | 2.0171 | 4.5149 | no-way (↔) |
| Spain | 6.558* | 2.2416 | one-way (→) |
| Sweden | 9.8445** | 4.5765 | one-way (→) |
| Turkey | 0.1987 | 5.5375 | no-way (↔) |
| United Kingdom | 2.9304 | 8.7291** | one-way (←) |

Notes:

*, denotes significance at the 10% level

**, denotes significance at the 5% level

***, denotes significance at the 1% level. DPGDP = first difference of PGDP; DDEPOP = second difference of the elderly population.

Source: Authors' illustrations based on STATA software.

examined objectively, it is clear that the ageing population significantly impacts the country's overall economic growth [57, 58].

It is evident that ageing will have a negative impact on economic growth. However, ageing is a natural part of the human experience. Therefore, the demographic dividend process will not last forever in a given economy. The movement or shift from one demography to another is referred to as demographic transition from a high birth and death rate to one that is low. In fact, unchecked population growth will result in a baby boom and/or population explosions. In the economy, neither situation is desirable. The economic scenario will follow a "family planning policy". It is worth noting that fertility, mortality, migration, and life expectancy are all factors that influence the ageing process in any country [59].

The aging population in Greece is putting significant strain on the country's social and health-care systems. The country should reform its social and health care policy agenda to address population ageing and its consequences. Further, these nations should implement fertility incentives and family policies to increase reproduction, as well as policies to improve the demographic structure and economic activity of migrants [60]. However, the unprecedented increase in ageing population in Europe poses new challenges to health, long-term care, and welfare systems. Regardless of potential morbidity compression, the number of older people with cancer, fractured hips, strokes, and dementia will increase, and many will have multiple morbidities. However, projected increases in health expenditure due to ageing are minor, and ageing does not pose a serious threat to the European welfare state. However, numerous options are available for adapting health care, long-term care, and welfare systems to better meet the needs of the ageing populations [61].

Implementing pragmatic policies is important at the micro and macro-economic level is paramount, which focuses on economic growth whilst addressing the problems of the ageing population [25]. The ageing population creates challenges and impacts the overall pension system, transportation, housing, health care, leisure, entertainment and even urban growth [9]. Therefore, no bidirectional causality relation is observed for any country and no causality relation is observed for eight countries (Belgium, Finland, France, Italy, Netherlands, Norway, Portugal, and Turkey), among the 15 countries in our dataset. Moreover, one-way causality relation is observed for seven countries: from elderly population growth rate to PGDP for two countries (Greece and the United Kingdom) and in the opposite direction for five countries (Luxembourg, Austria, Denmark, Spain, and Sweden).

Consequently, when taking control of these economic challenges, it is essential that the government, as the highest authority to make decisions, create solutions according to the context. The importance of doing so is that these government interventions can impact many different levels of the organisations, and even macro and micro components of a functional economy.

This analysis validates the growing burden of the ageing population in European economies, where almost three decades of consecutive budget deficits, gross government debt more than doubled to over 250% of GDP in 2020 [9]. The advancements in technology have been able to offset some of these ageing implications through the stimulus to increase the aggregate demand–yet the issue stems, as depicted above, from the reorientation of expenditures towards higher age-related spending, amid a declining tax base [9]. Especially in Europe, public social spending escalated from about 11% of GDP in the 1990s to about 22% of GDP in 2018, where 80% of the spending has been attributed to health care and long-term care on pensions for the ageing population [62]. Considering the current demographic trends are expected to worsen, it can further increase obligations on the society, i.e., when the ageing population grows, the majority of public expenditure will be drawn towards social expenditure related to the elderly, inevitably not providing a strong return–which was evident through the analysis as well [63].

There are opportunities for increasing output per capita as a result of the demographic "dividend" for two reasons. The ratio of "producers" to "consumers" increases as the proportion of people in working age to the overall population rises. This has two effects, undoubtedly having a positive impact on the rise in output per person. Second, "behavioural effects" on the growth of output per capita may also be present [64]. In countries such as Greece, United Kingdom etc., there is a substantial working elderly population. In any country, factors such as fertility, mortality, migration, and life expectancy have an impact on the ageing process [59]. There is no significant impact or change in the rate of economic growth when these factors occur in a slight and typical way. In contrast, an increase in the elderly population reduces economic growth because it dilutes capital [64]. The main policy consequences show that reducing government support for health care plans, pension plans and investing pension funds in high yielding mutual and bond markets can increase the return on investments made by the government on the elderly population. Government supply- or demand-side policies, as well as external shocks prevent the implementation of such policies for the elderly population, which lessens the impact on the economy.

## Wavelet coherence technique

The wavelet coherence approach was initially used in the engineering discipline but expanded to social sciences and other areas with time. It is a strong technique that could depict the direction and the leading variable between the two variables over different sections of time through the direction of the arrows. Rightward up means the second variable causes the first variable (positively), rightward down means the first variable causes the second variable (positively), leftward up means the second variable causes the first variable (negatively), and leftward down means the first variable causes the second variable (negatively). In this study, the first variable was economic growth, and the second variable was the elderly population. Furthermore, arrows appearing between 0.0 and 0.3 in the secondary vertical axis represent low frequency corresponding to long term. Arrows between 0.3 and 0.7 represent medium frequency corresponding to medium term and finally, 0.7 to 1.0 represent high frequency corresponding to short term. Thus, arrows pointing to different directions in a single period can exist. The graph was developed using R studio.

Fig 4 illustrates that the elderly population negatively led the economic growth, particularly in the 1960s and 1970s, as depicted by the down-left arrows. However, towards the latter part of the 20th century, particularly in the 1980s and 1990s, economic growth has been negatively led by the elderly population, as shown through the up-left arrows. In summary, the wavelet coherence technique indicates that initially, the elderly population was the dominant variable, but the directionality changed with time. It supplements Granger causality results, because in the 15 European countries considered, no country had shown bidirectionality, but seven countries had shown one-way directionality. Out of these seven countries, two showed the directionality from the elderly population to economic growth, while the rest of them showed the opposite direction. Perceptibly, the results of Granger causality are in alignment with the results of the wavelet coherence.

## Conclusions

The above-mentioned analysis indicates that the European region requires to strengthen the existing systems and adopt a well-planned approach to care for its ageing population. Nonetheless, these nations seem to have focused extensively on creating a strong foundation through caring for the ageing population and also minimising the ageing population's impact on the PGDP. In other words, opportunity costs like compromising investment on

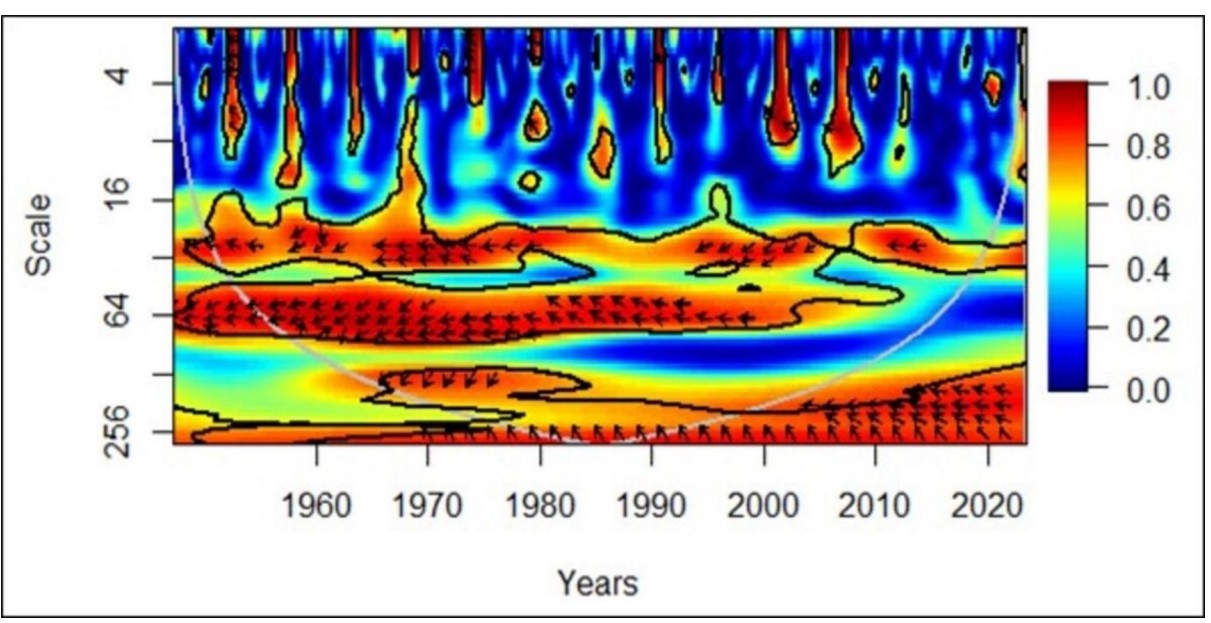

**Fig 4. European region: PGDP vs EPOP.** Source: Authors' illustrations based on R software.

development activities will be minimal. Nevertheless, the additional insights from the wavelet coherence approach show that the dominant variable (elderly population) negatively leads the other variable (economic growth) during most periods. Interestingly, these finding signal that neither variable is less important to ignore.

Accordingly, the strategic concentration should also be developed on the overall demographic trends of Europe, which is facing a sharp decline in population growth, yet with the rapid growth of the ageing population. As mentioned above, the intense impact on productivity drops, and the lack of facilities to manage a vast number of aged populations in crises such as COVID-19 pandemic are still lacking. Therefore, nations are in a highly vulnerable position. However, the impact of the ageing population on economic growth is minimal due to the government's predetermined policy planning on using the ageing population as a catalyst for growth, and several institutionalised cultural implications being developed. This reduces the burden on the economically active population. Yet, the rising longevity of human life will have to be addressed in several different contexts, where the burden lessens and becomes a long-term investment in society.

## Discussion and policy implications

The findings revealed that solutions necessitate improving the quality of living and reproductive capacity of individuals for a demographic shift in order to lessen the negative economic impact. From an economic perspective, the initial focus should be concentrated on the improvement of convenient types of employment, self-employment, and an infrastructure that supports its strategy, technologies and automation for labour-saving, as well as high school's early career guidance. On the other hand, it is important to continue to create conditions that will encourage pensioners to return to work. In terms of policy, it is required to gather and publish a broad range of socio-economic indicators, to connect the objectives, methodologies, and means of various programme material, and to provide additional workshops to city

workers on a regular basis as part of their career improvement. It will increase the effectiveness of the country's socioeconomic operational system.

The focus of policies for Europe should primarily be on organisations making extensive efforts to understand and re-evaluate the long-term pension plans and its challenges on the economy. More importantly, in 2007, pensions expansion was a burden for Greece, which was a contributory factor in the economic crisis [12]. Apart from Greece, Italy too has well-planned government-funded pension plans to provide social benefits for the majority of the ageing population. Therefore, this can strongly impact GDP growth unless properly managed, i.e., high government subsidies lead to long-term negative impacts on real GDP growth decline. Specially, the overall challenges posed by the increase of ageing population care falls directly on the workforce of a population–who will have to expand their earning capacity–and also impairing the quality of life [57]. Consequently, the major policy implications illustrate that tightening government subsidies on pension plans and investing on pension funds on high yielding mutual and bond markets can generate a strong return for the government investments made on the elderly population. Also, with the declining birth rate–the government should encourage European citizens to embrace motherhood and fatherhood, and parenthood in general. The Italians and Greeks tend to show strong tendency towards increasing child-births–while in contrast, the rest of Europe, especially the Nordic countries, tend to expect an average annual birth rate below 1% [63]. At least in professions that are less physically demanding, and in instances where policies are based on older workers' needs, working until an older age can maintain social integration and self-esteem. Employment and retirement policies are the result of many factors, and can help to make public expenditure sustainable, and can contribute to healthy ageing [61]. Lack of social interaction puts older people at risk for cognitive decline as well as depression, both of which are severe and expensive disabling ailments. Older people may be able to continue working, perhaps part-time, with the help of new work arrangements (like those described for health workforces). However, reintegration to society can also be achieved through participation in communal activities such as volunteering with charitable or community organisations [61].

As mentioned earlier, most secondary data for the study were obtained from the online database of the World Bank. However, the study focused only on the period between 1961 and 2021, and as per data availability, data for only 15 nations were gathered. In addition, when using a large data set such as 60 cases, the change in variables associated with a pandemic of two or three years is not taken into consideration separately. Moreover, by implementing new concepts and approaches, the present study can serve as a springboard for future studies and the research strategies could to be expanded to other continents. In the current study, authors inferred these research results using only two variables, which provides a framework for future research in this field. Future researchers can consider adding other demographic and economic variables and conducting extensive analysis. Particularly, influential moderator variables, such as educational level, wealth, and medical care, could be incorporated into future studies. Making policy implications for each of the countries considered is also crucial. However, given the available resources, the researchers were only able to derive a limited number of policy implications, which have been discussed here. Future researchers could further develop the study and expand the research where the wavelet coherence approach could be conducted to find correlations between individual European countries to better understand the nature of the relationship on a much deeper level.

The paper provides generic policies for Europe; therefore, future studies could further focus on country specific detailed analysis to facilitate improvement of existing policies.

## Supporting information

**S1 Appendix. Data file.**
(XLSX)

**S2 Appendix. Dfuller drift test.**
(DOCX)

**S3 Appendix. Dfuller trend test.**
(DOCX)

**S4 Appendix. Stability test.**
(DOCX)

**S5 Appendix. Lag length criteria results.**
(DOCX)

**S6 Appendix. VAR estimation.**
(DOCX)

## Acknowledgments

The authors would like to thank Ms. Gayendri Karunarathne for proof-reading and editing this manuscript.

## Author Contributions

**Conceptualization:** Thaveesha Jayawardhana, Ruwan Jayathilaka.

**Data curation:** Thaveesha Jayawardhana, Thamasha Nimnadi, Sachini Anuththara, Ridhmi Karadanaarachchi.

**Formal analysis:** Thaveesha Jayawardhana, Ruwan Jayathilaka, Thamasha Nimnadi, Sachini Anuththara, Ridhmi Karadanaarachchi.

**Methodology:** Thaveesha Jayawardhana, Ruwan Jayathilaka.

**Software:** Thaveesha Jayawardhana, Sachini Anuththara.

**Supervision:** Ruwan Jayathilaka, Kethaka Galappaththi.

**Validation:** Thaveesha Jayawardhana, Ruwan Jayathilaka, Thanuja Dharmasena.

**Visualization:** Thaveesha Jayawardhana, Ruwan Jayathilaka, Thamasha Nimnadi.

**Writing – original draft:** Thaveesha Jayawardhana, Ruwan Jayathilaka, Thamasha Nimnadi, Sachini Anuththara, Kethaka Galappaththi, Thanuja Dharmasena.

**Writing – review & editing:** Ruwan Jayathilaka, Thanuja Dharmasena.

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
