## [Decision Letter · Decision Letter 0]

5 May 2023

PONE-D-23-08974Cost of ageing: The relationship between the elderly population and economic growth in the European contextPLOS ONE

Dear Authors,

Thank you for submitting your manuscript to PLOS ONE. After careful consideration, we feel that it has merit but does not fully meet PLOS ONE’s publication criteria as it currently stands. Therefore, we invite you to submit a revised version of the manuscript that addresses the points raised during the review process. Based on the reviewers' reports I recommend major revision. You should revise carefully your manuscript according to their suggestions and respond to each point in a response letter to the reviewers. Please also highlight in color all your addings.

We look forward to receiving your revised manuscript.

Kind regards,

Magdalena Radulescu

Academic Editor

PLOS ONE

Journal Requirements:

Reviewers' comments:

Reviewer's Responses to Questions

**Comments to the Author**

1. Is the manuscript technically sound, and do the data support the conclusions?

Reviewer #1: Yes

Reviewer #2: Yes

2. Has the statistical analysis been performed appropriately and rigorously? 

Reviewer #1: Yes

Reviewer #2: Yes

3. Have the authors made all data underlying the findings in their manuscript fully available?

Reviewer #1: Yes

Reviewer #2: Yes

4. Is the manuscript presented in an intelligible fashion and written in standard English?

Reviewer #1: Yes

Reviewer #2: Yes

5. Review Comments to the Author

Reviewer #1: A manuscript with a very interesting title however, the work needs refinement. Introduction. I recommend that the authors highlight what is novel and what gap is filled by the work in relation to studies, especially the latest ones. In the methodology, the authors did not justify why they selected respondents over 65 years of age for the study. The strength of the work is an interesting methodology. The weakness is the very poor review of the literature. This section needs a major rewording, especially for 2022 and 2023.

Reviewer #2: Dear Author/s,

Thank you for giving me the opportunity to read your paper. The paper “Cost of ageing: The relationship between the elderly population and economic growth in the European context” is interesting for journal readers. But following changes should be done before the consideration to improve the quality of the paper:

Title: make the title sharper

1. The introduction part of the study needs improvement and story flow and the authors need to give proper contributions to their study.

2. I noticed that the novelty of this paper is not described in detail. This should be put in the introduction section properly.

3. There is a need to do a more rigorous and systematic literature review. The authors should clearly mention the literature gap. https://doi.org/10.1080/13683500.2020.1816929;
https://doi.org/10.1007/s11356-021-16720-2;
https://doi.org/10.1007/s11356-022-19106-0;
https://doi.org/10.1007/s11356-023-25698-y;
https://doi.org/10.1080/15567249.2016.1263251;
https://doi.org/10.1016/j.eap.2021.01.015;
https://doi.org/10.3390/su13084333;
https://doi.org/10.1007/s11356-019-04514-6;
https://doi.org/10.1007/s11356-019-06276-7;
https://doi.org/10.1007/s13132-011-0075-2;
https://doi.org/10.1007/s11356-021-12993-9;
https://doi.org/10.1007/s11356-021-12637-y;
https://doi.org/10.1080/13683500.2022.2158787

4. I would like to suggest that authors should update the introduction, literature, and results part. Specifically, the latest research trends, and in order to highlight the academic frontier of the research, the references of the recent year need to be referenced.

5. How did the authors get from the theoretical model to the empirical one? Behind the model there need to be a complete and well-thought-out theoretical grounding. This part of the article shouldn't include any citations or references; rather, it should be structured according to the authors' reasoning. The empirical model will come when this part has been completed.

6. The authors have only presented the findings, with no explanation of their economic reasoning. Do these findings validate or disprove the current policy framework? Are any new policy measures planned as a result of the findings? Discussion of the findings, which is conspicuously absent here, is meant to spark debate on policy. If the results don't offer anything new in terms of theory or policy, then a simple comparison with the literature won't prove their originality.

7. It would be appropriate to indicate a sharper future research directions and limitations of this at the end of the conclusion section just before references.

8. Poor references format: authors should address this issue in the revised draft. Kindly follow the right style of citation (references) throughout the manuscript by checking the guidelines of journal or any previously published paper in the journal.

6. PLOS authors have the option to publish the peer review history of their article (what does this mean?). If published, this will include your full peer review and any attached files.

Reviewer #1: No

Reviewer #2: No

---

## [Author Response · Author response to Decision Letter 0]

16 May 2023

Point by point response to editor and reviewers

Dear editor and the reviewers,

We would like to express our profound appreciation to the editor and the reviewers for the valuable comments and suggestions made on our manuscript. Your input has been incredibly helpful in revising and improving the quality of our work.

Please note that the line numbers referred to in this document are aligned with the revised manuscript which includes track changes.

Thank you once again for your valuable feedback.

Reviewer 1 comment 1: A manuscript with a very interesting title however, the work needs refinement. Introduction. I recommend that the authors highlight what is novel and what gap is filled by the work in relation to studies, especially the latest ones. In the methodology, the authors did not justify why they selected respondents over 65 years of age for the study. The strength of the work is an interesting methodology. The weakness is the very poor review of the literature. This section needs a major rewording, especially for 2022 and 2023.

Authors’ Response to Reviewer 1 comment 1: 

I highly appreciate your detailed comment, and it has been well noted. The changes you suggested have been incorporated into the revised manuscript, specifically highlighting the novelty of the study and addressing the research gap in line numbers 51-55, 97-100, and 110-112.

Furthermore, the literature has been updated to include articles from previous years, as indicated in line numbers 124-128 and 188-222.

Additionally, we have provided an explanation for why we selected the age of 65 years for the study, as mentioned in line numbers 242-243.

“….Economists have debated the connection between the ageing population and economic growth for an extended period. Researchers have disagreed about the relationship between the ageing population and economic expansion. Governments and professionals worldwide are interested in the impact of the former on economic growth, however, different age groups have varying degrees of productivity and, financial needs[1]…”

“…As such, the present research differs from the existing studies and fills a gap in the literature in many ways. Firstly, this study employs the latest data amassed over a considerable period and spans several nations in the European continent….”

“…. In addition, this research contributes to filling the empirical gap of capturing the Granger causality concerning the relationship between the elderly population and economic growth in the European context in a single study…..”

“…. For several decades, countries in Europe have seen an increase in elderly population. For example, in the European Union (EU), the old-age dependency ratio (the ratio of the population aged 65 or over to the population aged 15-64) nearly doubled from 15.2% in 1960 to 29.9% in 2016. Maintaining fiscal sustainability will be extremely difficult in the face of such a significant demographic shift [13]….”

“…..The global elderly population is growing as health and treatment standards improve, a phenomenon known as the “demographic revolution.” According to WHO, the global the elderly population is expected to reach 727 million by 2020 and more than 1.5 billion by 2050 [26]. The “demographic transition” initially resulted in unprecedented global population growth in the early 1970s. In general, rapid population growth contributed to swift increase in the number of workers, and fast increase in the number of workers contributed to speedy increase in GDP. As the demographic transition ends, the population is expected to grow slower and, after many decades, to reach a significantly high peak than currently [27] . Globally, an unprecedented shift in population demographics is taking place: for the first time, older people outnumber children. As the average age of the world's population rises, it not only poses problems for social welfare but also poses a significant challenge for energy systems and efforts to combat climate change. Europe, like the rest of the world, is undergoing a rapid demographic shift. Consistently low birth rates and higher life expectancy can explain the shift in the shape of the European Union's age pyramid [28]…..”

“…..Additionally, factors like the environment, transportation, and technology have an impact on how the economy is growing [29-35]. These factors are also influenced by the aging population. From the standpoint of public spending, when the relationship between economic growth and the environment is examined, it is emphasised that for both economic growth and a cleaner environment, governments should have balanced and sustainable public spending policies. Participation in environmental activities by the elderly population will boost economic growth to the extent that they value these factors. Economies that want to gain a competitive edge can raise their welfare level and achieve their goals in becoming information societies. Information technology has a short-term positive impact and a long-term negative impact on Turkey's economic growth [36]. By making the elderly population more familiar with technology and engaged in productive work, the country's economy might grow….”

“….Furthermore, when there is less risk and more safety protocols are in place, people who are concerned about public health would be kept at ease to attend the events. Event managers could benefit from a number of initiatives aimed at allaying consumer worries and fostering trust during the event in order to promote a secure and healthy atmosphere and encourage event participation [37]. Thereby, the mental health of elderly population will improve encouraging them to contribute more to the economy….” 

“….The elderly population is defined as people aged 65 and over [40]…..”

Reviewer 2 comment 1: Thank you for giving me the opportunity to read your paper. The paper “Cost of ageing: The relationship between the elderly population and economic growth in the European context” is interesting for journal readers. But following changes should be done before the consideration to improve the quality of the paper:

Title: make the title sharper

Authors’ Response to Reviewer 2 comment 1: 

Thank you for your valuable comment. It has been well noted and appreciated. 

Based on your suggestion, Title have been changed as “The cost of aging: Economic growth perspectives for Europe”.

Reviewer 2 comment 2: The introduction part of the study needs improvement and story flow and the authors need to give proper contributions to their study.

Authors’ Response to Reviewer 2 comment 2: 

Thank you for providing a detailed comment. Based on your feedback, the introduction has been adjusted to improve the story flow. The changes you suggested have been incorporated in line numbers 51-60. I appreciate your valuable input.

“…Economists have debated the connection between the ageing population and economic growth for an extended period. Researchers have disagreed about the relationship between the ageing population and economic expansion. Governments and professionals worldwide are interested in the impact of the former on economic growth, however, different age groups have varying degrees of productivity and, financial needs[1]. The number of persons aged 65 and above was expected to rise gradually over the world from 1960 to 2020. This number rose from 150 million in 1960 to 722 million in 2020 around the world….”

Reviewer 2 comment 3: I noticed that the novelty of this paper is not described in detail. This should be put in the introduction section properly.

Authors’ Response to Reviewer 2 comment 3: 

Thank you for your feedback. Your comment has been well received. Based on your suggestion, the novelty of this paper in the introduction has been adjusted. These changes have been incorporated into the revised manuscript, specifically in line numbers 51-60, 97-100, and 110-112. Thank you for your valuable input. 

“…Economists have debated the connection between the ageing population and economic growth for an extended period. Researchers have disagreed about the relationship between the ageing population and economic expansion. Governments and professionals worldwide are interested in the impact of the former on economic growth, however, different age groups have varying degrees of productivity and, financial needs[1]. The number of persons aged 65 and above was expected to rise gradually over the world from 1960 to 2020. This number rose from 150 million in 1960 to 722 million in 2020 around the world….”

“…. As such, the present research differs from the existing studies and fills a gap in the literature that in many ways. Firstly, this study employs the latest data amassed over a considerable period and spans several nations in the European continent….”

“….In addition, this research contributes to filling the empirical gap of capturing the Granger causality concerning the relationship between the elderly population and economic growth in the European context in a single study….”

Reviewer 2 comment 4: There is a need to do a more rigorous and systematic literature review. The authors should clearly mention the literature gap.

https://doi.org/10.1080/13683500.2020.1816929;
https://doi.org/10.1007/s11356-021-16720-2;
https://doi.org/10.1007/s11356-022-19106-0;
https://doi.org/10.1007/s11356-023-25698-y;
https://doi.org/10.1080/15567249.2016.1263251;
https://doi.org/10.1016/j.eap.2021.01.015;
https://doi.org/10.3390/su13084333;
https://doi.org/10.1007/s11356-019-04514-6;
https://doi.org/10.1007/s11356-019-06276-7;
https://doi.org/10.1007/s13132-011-0075-2;
https://doi.org/10.1007/s11356-021-12993-9;
https://doi.org/10.1007/s11356-021-12637-y;
https://doi.org/10.1080/13683500.2022.2158787

Authors’ Response to Reviewer 2 comment 4: 

Thank you very much for your comment and suggestions. I greatly appreciate it, and I have taken note of them. Those suggestions have been addressed in the literature review, specifically in line numbers 188-222. Thank you for your valuable input.

“…..The global elderly population is growing as health and treatment standards improve, a phenomenon known as the “demographic revolution.” According to WHO, the global the elderly population is expected to reach 727 million by 2020 and more than 1.5 billion by 2050 [26]. The “demographic transition” initially resulted in unprecedented global population growth in the early 1970s. In general, rapid population growth contributed to swift increase in the number of workers, and fast increase in the number of workers contributed to speedy increase in GDP. As the demographic transition ends, the population is expected to grow slower and, after many decades, to reach a significantly high peak than currently [27] . Globally, an unprecedented shift in population demographics is taking place: for the first time, older people outnumber children. As the average age of the world's population rises, it not only poses problems for social welfare but also poses a significant challenge for energy systems and efforts to combat climate change. Europe, like the rest of the world, is undergoing a rapid demographic shift. Consistently low birth rates and higher life expectancy can explain the shift in the shape of the European Union's age pyramid [28]….”

“…..Additionally, factors like the environment, transportation, and technology have an impact on how the economy is growing [29-35]. These factors are also influenced by the aging population. From the standpoint of public spending, when the relationship between economic growth and the environment is examined, it is emphasised that for both economic growth and a cleaner environment, governments should have balanced and sustainable public spending policies. Participation in environmental activities by the elderly population will boost economic growth to the extent that they value these factors. Economies that want to gain a competitive edge can raise their welfare level and achieve their goals in becoming information societies. Information technology has a short-term positive impact and a long-term negative impact on Turkey's economic growth [36]. By making the elderly population more familiar with technology and engaged in productive work, the country's economy might grow…..”

“….Furthermore, when there is less risk and more safety protocols are in place, people who are concerned about public health would be kept at ease to attend the events. Event managers could benefit from a number of initiatives aimed at allaying consumer worries and fostering trust during the event in order to promote a secure and healthy atmosphere and encourage event participation [37]. Thereby, the mental health of elderly population will improve encouraging them to contribute more to the economy…..” 

Reviewer 2 comment 5: I would like to suggest that authors should update the introduction, literature, and results part. Specifically, the latest research trends, and in order to highlight the academic frontier of the research, the references of the recent year need to be referenced.

Authors’ Response to Reviewer 2 comment 5: 

Thank you for your valuable comment. It has been well noted and appreciated. Based on your suggestions, the introduction, literature, and results sections have been updated. Additionally, the literature now includes articles from recent years. Thank you for your input.

Reviewer 2 comment 6: How did the authors get from the theoretical model to the empirical one? Behind the model there need to be a complete and well-thought-out theoretical grounding. This part of the article shouldn't include any citations or references; rather, it should be structured according to the authors' reasoning. The empirical model will come when this part has been completed.

Authors’ Response to Reviewer 2 comment 6: 

Comment well noted with thanks.

This study fills an empirical gap by capturing the Granger causality regarding the relationship between the elderly population and economic growth in the European context in a single study.

The Granger causality approach was used to examine the structures of the causal relationships between variables. The Granger causality test is a statistical hypothesis test used to determine whether one time series can forecast another.

Accordingly, the Methodology section has been updated with relevant citations. These updates have been incorporated in line numbers 257-261.

Thank you for your valuable input.

“…. Two stationary covariance variables, X and Y, are tracked over t periods. If it can be demonstrated that lagged values of a variable X have a significant influence on a regression model of Y that depends not only on X but also on its own lagged values Yt-1, Yt-2. Therefore, it can be argued that X Granger causes Y and potentially Y Granger causes changes in X [41]…..”

Reviewer 2 comment 7: The authors have only presented the findings, with no explanation of their economic reasoning. Do these findings validate or disprove the current policy framework? Are any new policy measures planned as a result of the findings? Discussion of the findings, which is conspicuously absent here, is meant to spark debate on policy. If the results don't offer anything new in terms of theory or policy, then a simple comparison with the literature won't prove their originality

Authors’ Response to Reviewer 2 comment 7:

Thanks for the detailed comment and suggestions.

Accordingly, the discussion and policy implication parts have been added and updated. The revised manuscript has addressed the comment in line numbers 553-564.

Thank you for your valuable input.

“….The findings revealed that solutions necessitate improving the quality of living and reproductive capacity of individuals for a demographic shift in order to lessen the negative economic impact. From an economic perspective, the initial focus should be concentrated on the improvement of convenient types of employment, self-employment, and an infrastructure that supports its strategy, technologies and automation for labour-saving, as well as high school’s early career guidance. On the other hand, it is important to continue to create conditions that will encourage pensioners to return to work. In terms of policy, it is required to gather and publish a broad range of socio-economic indicators, to connect the objectives, methodologies, and means of various programme material, and to provide additional workshops to city workers on a regular basis as part of their career improvement. It will increase the effectiveness of the country's socioeconomic operational system…..”

Reviewer 2 comment 8: It would be appropriate to indicate a sharper future research directions and limitations of this at the end of the conclusion section just before references.

Authors’ Response to Reviewer 2 comment 8:

 Well noted, and thank you for your comment.

The limitations and future studies have already been mentioned in line numbers 597-616. Thank you for your valuable input.

“…As mentioned earlier, most secondary data for the study were obtained from the online database of the World Bank. However, the study focused only on the period between 1961 and 2021, and as per data availability, data for only 15 nations were gathered. In addition, when using a large data set such as 60 cases, the change in variables associated with a pandemic of two or three years is not taken into consideration separately. Moreover, by implementing new concepts and approaches, the present study can serve as a springboard for future studies and the research strategies could to be expanded to other continents. In the current study, authors inferred these research results using only two variables, which provides a framework for future research in this field. Future researchers can consider adding other demographic and economic variables and conducting extensive analysis. Particularly, influential moderator variables, such as educational level, wealth, and medical care, could be incorporated into future studies. Making policy implications for each of the countries considered is also crucial. However, given the available resources, the researchers were only able to derive a limited number of policy implications, which have been discussed here. Future researchers could further develop the study and expand the research where the wavelet coherence approach could be conducted to find correlations between individual European countries to better understand the nature of the relationship on a much deeper level.

The paper provides generic policies for Europe; therefore, future studies could further focus on country specific detailed analysis to facilitate improvement of existing policies….”

Reviewer 2 comment 9: Poor references format: authors should address this issue in the revised draft. Kindly follow the right style of citation (references) throughout the manuscript by checking the guidelines of journal or any previously published paper in the journal.

Authors’ Response to Reviewer 2 comment 9: 

Noted with thanks! All the references have now been arranged according to the guidelines. Thank you for your valuable input.

---

## [Decision Letter · Decision Letter 1]

1 Jun 2023

The cost of aging: Economic growth perspectives for Europe

PONE-D-23-08974R1

Dear Dr. Jayathilaka,

We’re pleased to inform you that your manuscript has been judged scientifically suitable for publication and will be formally accepted for publication once it meets all outstanding technical requirements.

Kind regards,

Magdalena Radulescu

Academic Editor

PLOS ONE

Additional Editor Comments (optional):

Dear Authors,

Based on reviewers' reports, the paper can be accepeted.

Best regards,

Magdalena Radulescu

Associate Editor, PlosOne

Reviewers' comments:

Reviewer's Responses to Questions

**Comments to the Author**

1. If the authors have adequately addressed your comments raised in a previous round of review and you feel that this manuscript is now acceptable for publication, you may indicate that here to bypass the “Comments to the Author” section, enter your conflict of interest statement in the “Confidential to Editor” section, and submit your "Accept" recommendation.

Reviewer #1: All comments have been addressed

Reviewer #2: All comments have been addressed

2. Is the manuscript technically sound, and do the data support the conclusions?

Reviewer #1: Yes

Reviewer #2: Yes

3. Has the statistical analysis been performed appropriately and rigorously? 

Reviewer #1: Yes

Reviewer #2: Yes

4. Have the authors made all data underlying the findings in their manuscript fully available?

Reviewer #1: Yes

Reviewer #2: Yes

5. Is the manuscript presented in an intelligible fashion and written in standard English?

Reviewer #1: Yes

Reviewer #2: Yes

6. Review Comments to the Author

Reviewer #1: Manuscript has been improved accordance with recommendations. The authors referred to all the comments that were included in the review. The article is readable, well-designed, and interesting.

Reviewer #2: Thank you for giving me the opportunity to read your paper.

Thank you for giving me the opportunity to read your paper.

7. PLOS authors have the option to publish the peer review history of their article (what does this mean?). If published, this will include your full peer review and any attached files.

Reviewer #1: No

Reviewer #2: No

---

## [Editor Report · Acceptance letter]

13 Jun 2023

PONE-D-23-08974R1 

The cost of aging: Economic growth perspectives for Europe 

Dear Dr. Jayathilaka:

I'm pleased to inform you that your manuscript has been deemed suitable for publication in PLOS ONE. Congratulations! Your manuscript is now with our production department. 

Kind regards, 

on behalf of

Dr. Magdalena Radulescu 

Academic Editor

PLOS ONE